# Cross-serotypically conserved epitope recommendations for a universal T cell-based dengue vaccine

**Syed Faraz Ahmed**[1], **Ahmed A. Quadeer**[1]*, **John P. Barton**[2]*, **Matthew R. McKay**[1,3]*

**1** Department of Electronic and Computer Engineering, The Hong Kong University of Science and Technology, Hong Kong, China, **2** Department of Physics and Astronomy, University of California, Riverside, California, United States of America, **3** Department of Chemical and Biological Engineering, The Hong Kong University of Science and Technology, Hong Kong, China

* eeaaquadeer@ust.hk (AAQ); john.barton@ucr.edu (JPB); m.mckay@ust.hk (MRM)

**Data Availability Statement:** All sequence data files are available from the ViPR database. Unique identifiers of all protein sequences as well as all epitope sequences are listed in the Supporting

## Abstract

Dengue virus (DENV)-associated disease is a growing threat to public health across the globe. Co-circulating as four different serotypes, DENV poses a unique challenge for vaccine design as immunity to one serotype predisposes a person to severe and potentially lethal disease upon infection from other serotypes. Recent experimental studies suggest that an effective vaccine against DENV should elicit a strong T cell response against all serotypes, which could be achieved by directing T cell responses toward cross-serotypically conserved epitopes while avoiding serotype-specific ones. Here, we used experimentally-determined DENV T cell epitopes and patient-derived DENV sequences to assess the cross-serotypic variability of the epitopes. We reveal a distinct near-binary pattern of epitope conservation across serotypes for a large number of DENV epitopes. Based on the conservation profile, we identify a set of 55 epitopes that are highly conserved in at least 3 serotypes. Most of the highly conserved epitopes lie in functionally important regions of DENV non-structural proteins. By considering the global distribution of human leukocyte antigen (HLA) alleles associated with these DENV epitopes, we identify a potentially robust subset of HLA class I and class II restricted epitopes that can serve as targets for a universal T cell-based vaccine against DENV while covering ~99% of the global population.

## Author summary

The rise in global incidence of DENV and the resulting rise in mortality rate necessitates an effective universal vaccine against it. Since infection from one DENV serotype makes a person vulnerable to severe disease upon infection from another serotype, an effective vaccine should protect against all DENV serotypes. Increasing experimental evidence suggests that T cells are important for protecting against DENV. In this work, we comprehensively analyzed the extensive publicly-available data on DENV and revealed a distinct pattern of epitope conservation for several of the DENV T cell epitopes. Importantly, we identified a set of epitopes that are highly conserved across at least three DENV

Information tables. All source code and scripts for reproducing the results are available at https://github.com/faraz107/Robust-DENV-Vaccine-Candidates.

**Funding:** This work was supported by the General Research Fund of the Hong Kong Research Grants Council (RGC) (grant numbers 16204519 and 16201620). SFA was also supported by the Hong Kong Ph.D. Fellowship Scheme (HKPFS) (number PF15-10255). The funders had no role in study design, data collection and analysis, decision to publish, or preparation of the manuscript.

**Competing interests:** The authors have declared that no competing interests exist.

serotypes. Incorporating information about the HLA alleles associated with these epitopes, we identified a potentially robust set of epitopes as targets for a prospective universal T cell-based vaccine that maximizes global population coverage. By training the immune system to target specific regions of DENV proteins which are likely to protect against multiple serotypes, a T cell-based vaccine might be effective in circumventing progression to severe dengue infection.

## Introduction

Dengue is a mosquito-borne viral infection that mostly causes febrile illness, but in some cases progresses into a life-threatening disease [1]. It is caused by the dengue virus (DENV) which exists as four genetically and antigenically distinct serotypes [2,3]. Global incidence of dengue has increased more than 6-fold over the last two decades [4] with an estimated 400 million new infections occurring annually [5]. Dengue is now endemic in at least 100 countries with multiple DENV serotypes co-circulating in various parts of the world [6] and it is estimated that 3.9 billion people are at risk of being infected by DENV [7]. Despite several efforts, there is currently no safe and effective vaccine that protects against dengue [8]. The only licensed dengue vaccine, Dengvaxia, does not completely protect against all DENV serotypes and, more importantly, increases the risk of severe disease among young children who are already most vulnerable to DENV infections [9,10].

One of the main hindrances in the development of an effective DENV vaccine is its complicated pathogenesis: primary DENV infections are mostly asymptomatic but secondary infection with a different serotype (i.e., heterologous infection) can cause clinical complications and lead to severe forms of disease [3]. Severe dengue has been associated with immunopathology where an aberrant immune response exacerbates the disease outcome [11]. Although both arms of the adaptive immune response—antibodies and T cells—have been investigated, the sub-neutralizing antibodies seem to play a major role in progression to severe dengue during secondary infection via a phenomenon known as antibody-dependent enhancement (ADE) of disease [12,13]. In contrast, there is no explicit evidence for the role of T cells in progression to severe dengue as of yet [14,15]. In fact, increasing evidence suggests a protective role of T cells against DENV infection [13–18]. Moreover, ADE as well as the lack of a DENV-specific T cell response following Dengvaxia vaccination have been suggested to be the main factors leading to an increased risk of severe dengue among vaccine recipients [10]. These recent developments suggest the potential importance of a vaccine designed to induce protective T cell responses.

Most of the vaccines against DENV that are currently in advanced stages of development are based on live-attenuated viruses (LAVs) comprising variant chimeric forms of DENV serotypes [19–22]. These vaccines are expected to induce strong T cell responses against the vaccine-strain-specific immunodominant epitopes, but may not be protective against multiple DENV serotypes [23–26]. The immunodominance hierarchy of DENV T cell epitopes is complex and has been shown to depend upon the serotypes of current and past infecting strains. That is, for primary and subsequent homologous infections, T cell responses are dominant towards serotype-specific epitopes, while for secondary heterologous infections the responses skew towards epitopes that are conserved across multiple DENV serotypes [27–29]. This expansion of T cell responses towards conserved epitopes following heterologous secondary infection is thought to play a protective role, resulting in a low incidence rate of symptomatic tertiary infections [28,30]. Notably, cross-reactive CD8+ T cells that target conserved epitopes

in multiple DENV serotypes have been shown to mediate protection in heterologous secondary infections [14,15,31,32]. Experiments using mice have shown that cross-reactive CD8+ T cells provide protection from DENV even in the presence of disease-enhancing antibodies [33,34]. These recent findings strongly suggest that a vaccine eliciting a T cell response specifically targeting epitopes conserved across serotypes is likely to enable protection against DENV, as highlighted in a recent review [13].

Thus far, only a few studies have attempted to identify protein regions (peptides) conserved across multiple DENV serotypes that may be targeted by T cell-based vaccines [18,35–37]. Specifically, in one such study [35], conservation of overlapping peptides across sequence data of multiple serotypes was used to identify 44 pan-DENV conserved peptides of varying lengths (9 to 22 residues). In another study [36], a block entropy based method was employed to identify 1,551 pan-DENV conserved blocks of 9-mer peptides from sequence data available at the time of study. The selection of conserved peptides in this study was not strict as it allowed for variants (i.e., peptides that differ from each other at few residues either within a single serotype or across serotypes). Similar to [36], studies [37] and [18] allowed for variants and reported 46 and 11 conserved epitopes, respectively. Although all these studies [18,35–37] reported multiple conserved regions/epitopes within the DENV proteome, they did not consider other factors that are important for the design of a cross-serotypic dengue vaccine. First, the conserved peptides/blocks predicted exclusively from DENV sequences (as in [35,36]) may not represent T cell epitopes that are actually presented on HLA molecules of DENV-infected cells. Second, training the immune system using variants of peptides/epitopes [18,36,37], especially those which are serotype-specific, may result in sub-optimal T cell responses in subsequent heterotypic infections [27]. These issues may be addressed respectively by exclusively using data on experimentally-determined DENV T cell epitopes (a large number of which have been determined over the past few years), and by selecting conserved epitopes based on a strict criterion that discounts variants. Thus, using the available immunological data, together with a strict approach to determine epitope conservation (i.e., which considers each variant as a separate epitope) based on the publicly-available patient-derived DENV sequence data, could enable the identification of cross-serotypically conserved T cell epitopes that can aid the rational design of a potentially effective universal vaccine against dengue.

Here, we take an epitope-centric approach by focusing on all available data of experimentally-determined DENV T cell epitopes that are derived from human hosts and carry associated HLA information. This makes our approach distinct from previous related works as the T cell epitopes that we consider are not only known to be processed and presented by the infected cells, but they also elicit positive/protective T cell responses within humans in the context of cognate HLA alleles. By examining the conservation of each available T cell epitope in the protein sequence data for each DENV serotype, our analysis reveals a striking near-binary pattern of conservation across serotypes for multiple epitopes. That is, epitopes are often highly conserved in particular serotypes and nearly absent from the remaining ones. We employ a conservative approach for mapping the epitopes onto sequences, which excludes non-conserved epitope variants from the analysis. This is important as mutation of even a single epitope residue is known to potentially abrogate HLA-epitope binding and significantly weaken the T cell response [38–40]. Based on the determined conservation profile of the epitopes, we identify a set of 55 epitopes that are strictly conserved in at least three of the four DENV serotypes. The majority of these epitopes are derived from functionally important regions of DENV non-structural proteins (NS3 and NS5), highlighting their potential as effective vaccine targets. Incorporating the HLA information associated with the identified cross-serotypically conserved epitopes, we propose a set of robust epitopes as targets for a prospective T cell-based universal vaccine against DENV that can potentially protect a large

percentage of the global population. Moreover, to demonstrate the further applicability of our approach, we also propose country-specific vaccine targets for three dengue-endemic countries (Thailand, Brazil and Philippines), that are experiencing a significant recent rise in number of DENV infections [41].

## Materials and methods

### Data acquisition and processing

We downloaded all available DENV protein sequences derived from human hosts from the NIAID Virus Pathogen Resource (ViPR) database [42] (https://www.viprbrc.org/; accessed July 30, 2019). The downloaded sequences were aligned separately for each DENV protein (E, prM, C, NS1, NS2a, NS2b, NS3, NS4a, NS4b, and NS5) and each serotype (DENV1-4) using the MAFFT multiple sequence alignment program [43]. Sequences that had less than 15% gaps within the alignment were retained for further analysis, which resulted in a total of 56,496 protein sequences (Fig 1) (see S1 Table for accession numbers of all protein sequences). Most of these sequences are from DENV1 and DENV2. The fewest are from DENV4, for which reporting of sequences has increased the least rapidly over the past several years [6]. Among DENV proteins, a large number of envelope protein (E) sequences are reported as compared to other proteins. This is expected, with envelope being a major focus of dengue research as it is the main target of antibodies.

We also downloaded data of all available DENV T cell epitopes from the NIAID ViPR database [42] (https://www.viprbrc.org/; accessed November 15, 2019) that were derived from human hosts. This data comprises both HLA class I and class II restricted epitopes, including those associated with the recently characterized class II loci (HLA-DP, DQ, and DRB3/4/5) [44]. These were grouped according to protein by separately filtering the epitope data for each protein. For this, the starting and ending protein positions were defined based on the following reference sequences: NP_059433.1 (DENV1), NP_056776.2 (DENV2), YP_001621843.1 (DENV3) and NP_073286.1 (DENV4). Of all the obtained epitopes, we retained only those that were reported to be positive in at least one T cell assay, and for which HLA information was available. This procedure resulted in a total of 1,768 (674 HLA class I and 1,094 HLA class II restricted) DENV T cell epitopes (Fig 2A and S2 Table).

### Computing conservation of epitopes

We measured the conservation of each epitope for each serotype as the fraction of protein sequences within that serotype that comprised the exact "epitope sequence". That is, to compute the conservation of an epitope, we used an exact mapping procedure (Fig 2B) that screened each epitope against all protein sequences within each serotype and mapped the epitope onto a particular sequence only when it matched identically to the corresponding region of that sequence. The fraction of sequences to which the epitope was mapped represented its conservation in the respective serotype. This strict approach ensured that (variant) epitopes differing from each other at one or more residues were considered separately.

### Computing epitope coverage of DENV proteome

We defined the epitope coverage at any position of a protein as the number of epitopes that covered that position. To compute it, we first mapped all the T cell epitopes onto all sequences of the corresponding proteins for each serotype using the exact mapping procedure explained above (Fig 2B). The epitope coverage of a protein was then computed after accounting for all

**A**

| Protein | DENV1 | DENV2 | DENV3 | DENV4 | All serotypes |
|---------|-------|-------|-------|-------|---------------|
| E | 6680 | 4238 | 2187 | 1561 | 14666 |
| prM | 2177 | 1851 | 1048 | 360 | 5436 |
| C | 1999 | 1691 | 939 | 286 | 4915 |
| NS1 | 1938 | 1537 | 918 | 245 | 4638 |
| NS2a | 1893 | 1470 | 907 | 241 | 4511 |
| NS2b | 1896 | 1462 | 907 | 245 | 4510 |
| NS3 | 1876 | 1467 | 922 | 238 | 4503 |
| NS4a | 1871 | 1457 | 910 | 231 | 4469 |
| NS4b | 1872 | 1442 | 909 | 229 | 4452 |
| NS5 | 1852 | 1420 | 895 | 229 | 4396 |
| Total | 24054 | 18035 | 10542 | 3865 | 56496 |

**B**

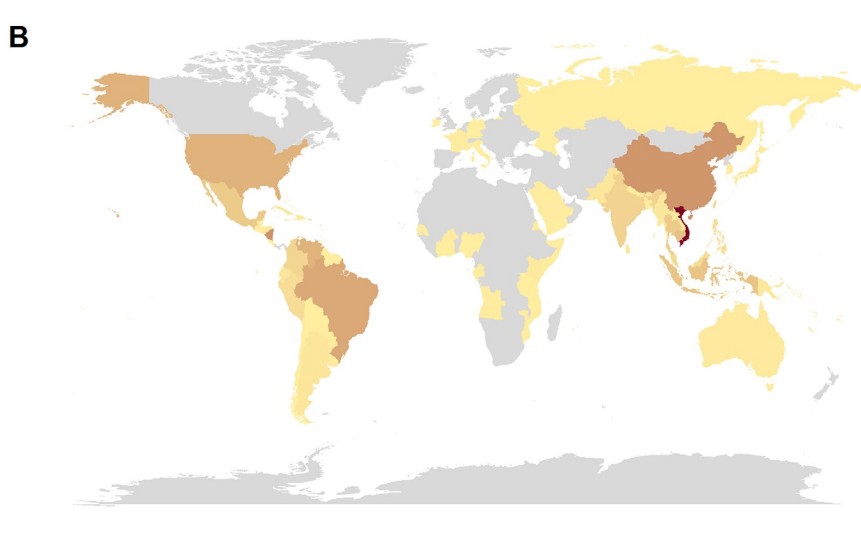

Number of DENV protein sequences

3000   6000   9000

**Fig 1. Analyzed DENV sequences span the DENV genome and come from multiple geographic locations. (A)** Number of DENV protein sequences for each of the four serotypes of DENV retrieved from the ViPR database [42]. **(B)** Locations where DENV protein sequences used in this study were sampled. Countries from where no sequence was available for analysis are colored as grey. The map was generated using the "maps" package (https://cran.r-project. org/web/packages/maps/index.html) through the R programming language.

epitopes that mapped onto at least one sequence of that protein. We found the coverage of DENV proteins to be qualitatively similar across all serotypes (Fig 2C, S1 Fig).

## Estimating population coverage of epitopes

Population coverage is an estimate of the percentage of individuals in a target population that are likely to be able to mount a T cell response against a given epitope. We computed the population coverage of a T cell epitope based on the associated HLA alleles in the ViPR database that were reported together with the positive T cell assay data. We only considered alleles with

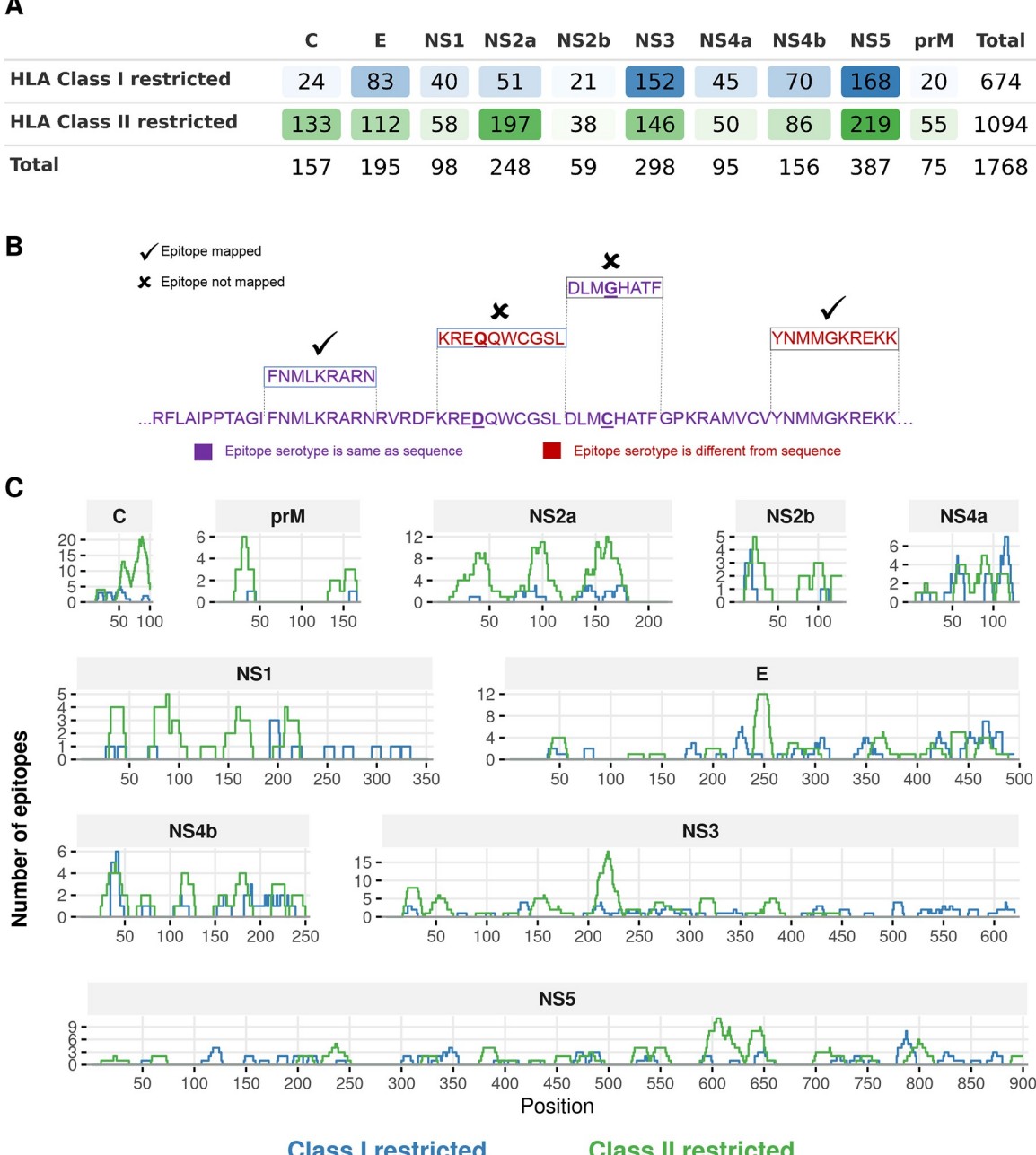

**Fig 2. Statistics and coverage of experimentally-determined DENV T cell epitopes. (A)** Number of DENV epitopes derived from different proteins across the serotypes, from the ViPR database [42]. **(B)** Procedure for the exact mapping of an epitope to a protein sequence. Epitopes were mapped onto the sequence only when all epitope residues exactly matched with the protein residues, i.e. no residue mismatches were allowed. **(C)** Coverage of T cell epitopes across DENV proteins. The locations of epitopes were determined by mapping them onto all DENV1 sequences. Similar patterns of epitope coverage were observed for most proteins when the epitopes were mapped onto sequences of DENV2/3/4 (see S1 Fig).

at least 4-digit resolution (e.g., A*02:01). The (accumulated) population coverage of a set of epitopes is defined as the percentage of individuals in a target population that are likely to be able to respond to at least one T cell epitope within the set. We adopted a greedy approach to identify the set of epitopes from among the highly conserved ones which maximized the

accumulated population coverage. The steps of the approach were as follows: (i) we initialized the set with the epitope that had the highest individual population coverage; (ii) added to the set another epitope which maximized the accumulated population coverage; and (iii) continued adding epitopes until we obtained a set of epitopes for which the accumulated population coverage did not increase further by adding any of the remaining highly conserved epitopes. If at any step the same increase in accumulated population coverage resulted from adding multiple epitopes, then only the most conserved epitope among them was added to the set.

The Python code of the tool for computing the population coverage was downloaded from the IEDB Analysis Resource [45] (http://tools.iedb.org/population/download/; accessed August 1, 2019) and run locally. This tool uses the population-specific HLA allele genotypic frequencies from the Allele Frequency database [46] (http://www.allelefrequencies.net/).

## Results

### Mapping T cell epitopes onto the DENV proteome reveals their broad and non-uniform coverage

We first mapped the downloaded experimentally-determined HLA class I and II restricted T cell epitopes (summarized in Fig 2A) onto the protein sequences for each DENV serotype by using an exact mapping criterion; i.e., no mismatch was allowed between the epitope sequence and the corresponding region of the protein sequence (Fig 2B). We then determined the coverage of the mapped epitopes in each DENV serotype (see Methods for details). We found that for most DENV proteins the epitope coverage was broad, with at least 70% of C, NS2a, NS3, NS4a and NS4b residues being covered by at least one T cell epitope (Fig 2C, S1 Fig). Among the remaining proteins, coverage of E, NS1, NS2b and NS5 was moderate (40–70%), while prM was the least covered (less than 40%). This suggests that all DENV proteins are targeted by natural T cell responses, and that a large part of the DENV proteome is potentially immunogenic.

The mapping of epitopes onto the DENV proteome further revealed that the epitope coverage is distributed non-uniformly across different proteomic regions (Fig 2C, S1 Fig). Interestingly, specific "immunologically dense" regions, defined as regions covered by multiple epitopes, were observed in some proteins, including C (residues 50–100), NS2a (residues 150–180) and NS3 (residues 200–250). Moreover, most of the protein regions were covered by both HLA class I and II restricted epitopes, while some were covered by only one type of epitope. For example, both HLA class I and class II restricted epitopes covered most regions of NS2a, NS4b and NS5 proteins, while most of the C-terminal region of NS3 was covered only by HLA class I restricted epitopes and that of the C protein was covered only by HLA class II restricted epitopes. Epitope coverage is important to understand the proteomic regions accessible to T cells. It is also important, however, to examine their conservation across DENV serotypes in order to identify specific cross-serotypically conserved epitopes that may serve as potentially effective targets for T cells.

### Conservation profiles of epitopes reveal a near-binary pattern across serotypes

In order to compute the conservation profiles of the T cell epitopes, we first mapped them onto all available protein sequences for each serotype (Methods). Similar to above, we used an exact mapping criterion such that no mismatch was allowed between the epitope and any protein sequence (Fig 2B). This conservative mapping precluded one- or multiple-residue variants of an epitope from spuriously inflating its conservation profile. This is important from the

perspective of identifying vaccine targets since single residue substitutions have been demonstrated to influence epitope presentation, weaken or completely abrogate epitope-HLA molecule binding, and result in insufficient T cell responses [38–40]. Furthermore, it is particularly important to avoid training the immune system against epitope variants in the case of DENV, as targeting these can lead to weakened T cell responses (i.e., T cell responses to the variant epitopes may be weakly cross-reactive), and such effects have been implicated in offering a lack of protection and progression to severe disease [31,47,48].

Following the exact mapping of epitopes, we found 13 highly conserved epitopes, representing less than 1% of all available DENV epitopes across all four DENV serotypes. All of these epitopes were derived from non-structural proteins, with the majority (8 out of 13) from NS5 (Fig 3). The conservation profiles of the remaining epitopes revealed a distinct pattern: a large number of epitopes were highly conserved in 3 serotypes and nearly absent (almost zero conservation) in the fourth serotype. This unique near-binary pattern apparently resulted from the presence of serotype-specific residues within these epitopes. For example, the NS5 epitope LEFEALGF**L** was highly conserved (>99%) within each of DENV2, DENV3 and DENV4 but absent in DENV1 where, in the last epitope position, Leucine (L) was substituted by Methionine (M). Similarly, the envelope epitope QEGAMH**T**AL was highly conserved in DENV1, DENV2 and DENV3 but not in DENV4 where the variant QEGAMH**S**AL was highly conserved (S2 Fig). This distinct pattern of epitope conservation extended beyond the top epitopes and several other epitopes were found to be highly conserved within only two or one DENV serotype while being absent from the remaining serotypes (S3 Fig).

To identify robust T cell targets against multiple DENV serotypes, we excluded epitopes that are highly conserved in only one or two serotypes (~42% and ~6% of all DENV epitopes respectively), and instead focused only on epitopes that are expected to be protective against a majority of serotypes. Based on the conservation profile, we identified a set of 55 DENV epitopes that had a minimum conservation level above a high threshold (0.9) within at least 3 of the 4 serotypes (Fig 3). This set of epitopes was found to be robust to the specific choice of threshold used for defining the minimum conservation level (S4 Fig).

While the identification of epitopes that are conserved across multiple serotypes is critical to select targets for a prospective vaccine, examining the immunodominance of these epitopes across multiple individuals is also important. Measuring the immunodominance of epitopes, in general, is complicated and varies across studies [48,49]. Here, we used two approaches to quantitatively compare the immunodominance of epitopes. First, we extracted from IEDB the information about the number of positive assays and donors reported for each of the top 55 conserved epitopes (Fig 3). Half of these epitopes (28/55) were reported positive in multiple assays while nearly two-thirds (35/55) were reported positive in multiple donors, suggesting that these can be considered as immunodominant epitopes [48]. Second, we compared the response frequencies reported for each of these epitopes in the IEDB database (Fig 3). The median of the response frequencies for these epitopes was 0.09. Comparing this against the distribution of the reported response frequencies for all DENV T cell epitopes in IEDB, we found that all 55 identified epitopes scored above ~43% of all DENV T cell epitopes (Fig 3). In addition, many of our identified epitopes were reported to have very strong affinities with cognate HLA molecules (S8 Fig and S2 File); a factor considered important for eliciting a strong T cell response [14]. Moreover, we found that several of our identified epitopes were listed as being immunodominant in multiple previous studies [26,31,48] (S3 Table).

So far, we have focused only on epitopes determined from human hosts. However, we can also use our framework to analyze all the (213) experimentally-determined T cell epitopes from HLA transgenic mice reported in the ViPR database [42] (S3 File). Using our strict criteria, we found only 9 epitopes from HLA transgenic mice to be conserved (conservation greater

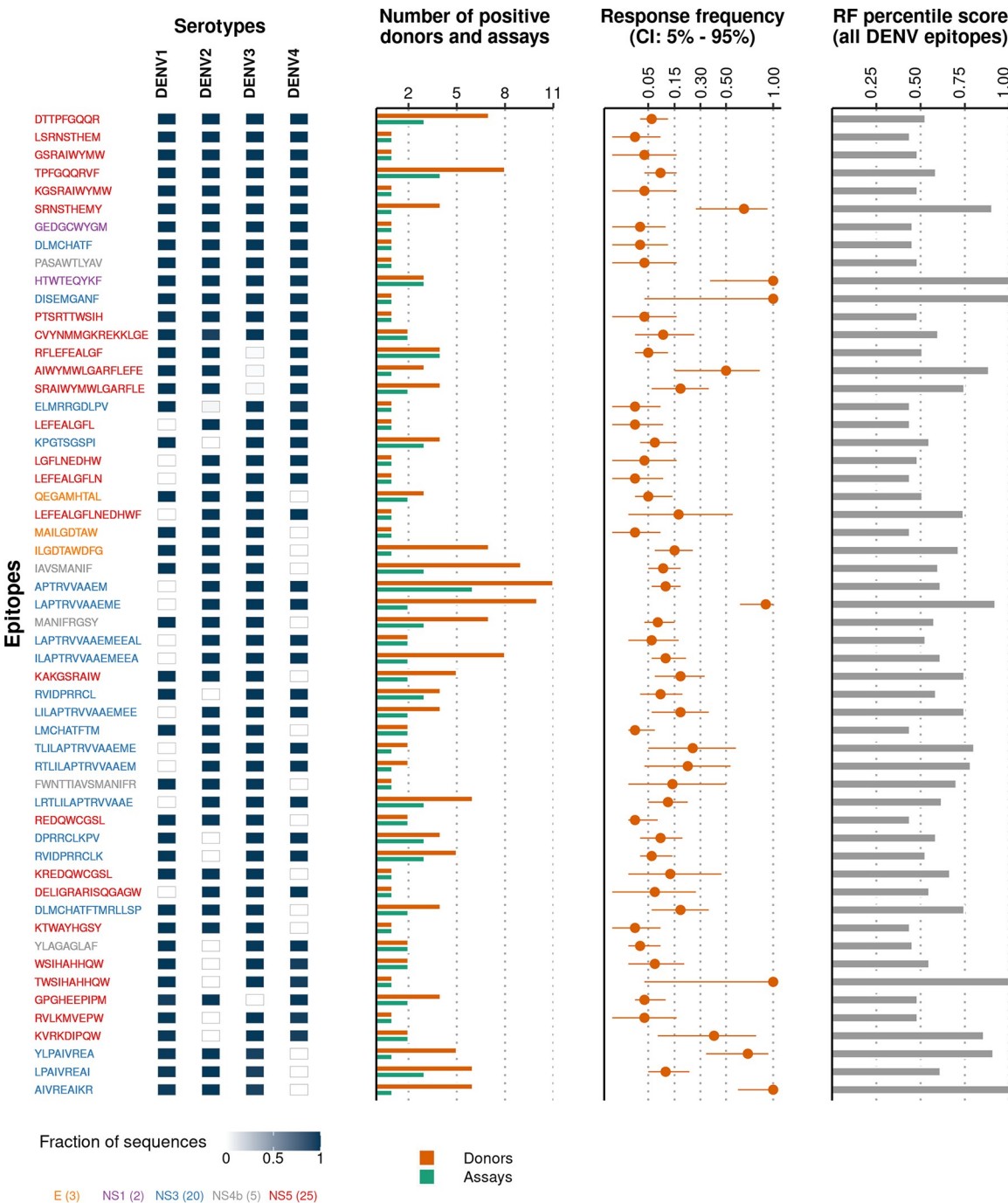

**Fig 3. Top cross-serotypically conserved DENV T cell epitopes exhibit near-binary conservation.** Of the 1,768 experimentally-determined epitopes, 55 mapped exactly onto at least 90% of sequences in at least 3 of the 4 DENV serotypes. The cells adjacent to each epitope represent its conservation within each DENV serotype. The number of donors and assays that have tested positive and the response frequency (RF) as reported in IEDB [45] are shown for each epitope. RF percentile score shows the response frequencies of each epitope compared to those reported for all DENV epitopes in IEDB [45] (S2 File). Epitopes are arranged according to their mean conservation (in descending order) and colored according to the protein from which they were derived. The number of epitopes within the set derived from each protein is shown within parentheses in the legend. For the conservation profile of all available (1,768) T cell epitopes, see S3 Fig.

than 0.9) in at least 3 serotypes (S9 Fig). Five of these epitopes (namely, APTRVVAAEM, ELMRRGDLPV, RVIDPRRCL, SRAIWYMWLGARFLE, and LPAIVREAI) were also reported from human hosts and are thus present in our top 55 conserved epitopes (Fig 3).

To complement these results, we developed a software tool that filters all available DENV epitopes based on any choice of threshold for the minimum conservation level of epitopes within each serotype. This easy-to-use tool presents the conservation profiles of all 1,768 experimentally-determined DENV T cell epitopes considered in the present work (see Methods) and is provided as a standalone HTML file (S1 File) which can also help to quickly identify the most conserved epitopes from any DENV protein. Thus, this tool may serve as a useful reference for future research on DENV epitopes.

## Top conserved DENV T cell epitopes in NS3 and NS5 are located in functionally important regions

Most of the identified conserved epitopes (45 out of 55) were derived from NS5 and NS3 (25 and 20, respectively) (Fig 3), highlighting the potential biological importance of targeting these non-structural proteins. Both NS5 and NS3 are large DENV proteins, comprising about 900 and 618 residues respectively (lengths slightly vary across serotypes), which are multifunctional enzymes playing a critical role in viral replication [50–52]. The high conservation of the identified NS5 and NS3 epitopes across serotypes (Fig 3) indicates that the protein regions associated with them may be structurally/functionally important for DENV. To investigate this, we mapped these epitopes to the available protein crystal structures of these proteins. These top conserved epitopes were found to lie deep within the protein tertiary structures and generally in the proximity of functionally important regions such as RNA binding sites and known flavivirus motifs.

Among the top conserved NS5 epitopes, two of them (SRNSTH**E**MY and LSRNSTH**E**M) contained an active site (E216) of the enzymatic tetrad of methyltransferase (MTAse) domain known to be critical for viral replication [53]. Four of the top conserved NS5 epitopes (DTTPFGQQR, TPFGQQRVF, KTWAYHGSY and GPGHEEPIPM) contained residues that are known to be involved in interactions that provide overall stability to the NS5 quaternary structure [54]. Some residues involved in these epitopes are known to be functionally critical as well [53]; for example, mutations in epitopes DTTP**F**GQQR (F349D) and SRNSTH**E**MY (E216A) were shown to severely impair viral replication. Four other epitopes among the top conserved NS5 epitopes (WSIHAHHQW, TWSIHAHHQW, PTSRTTWSIH and CVYNMMGKREKKLGE) were found to overlap with the priming loop, which is important for stabilizing the RNA-dependent RNA-polymerase (RdRp) initiation complex [52]; and motif F, which mediates conformational changes between the thumb and fingers domains of NS5 to provide stability to the RdRp domain [55] (Fig 4A). Furthermore, two of the top conserved epitopes of NS5 (GPGHEEPIPM and KVRKDIPQW) contained residues involved at the NS5 inter-dimer interface [53] (S5 Fig).

Among the top conserved NS3 epitopes, one epitope (KPGT**S**GSPI) comprised an active site (S135) of the catalytic triad, mutations at which have been shown to completely abolish the protease activity [56]. All remaining top conserved NS3 epitopes were found to lie within the conserved flavivirus motifs and map to the helicase domain [57]. Remarkably, while these NS3 epitopes were located far from each other in the primary structure (Fig 4B), they localized in the tertiary structure around the groove that is important for interacting with viral RNA and facilitating its unwinding [50,58].

Overall, this analysis suggested that the identified top conserved epitopes of NS5 and NS3 are derived from functionally/structurally important regions of these proteins. These results

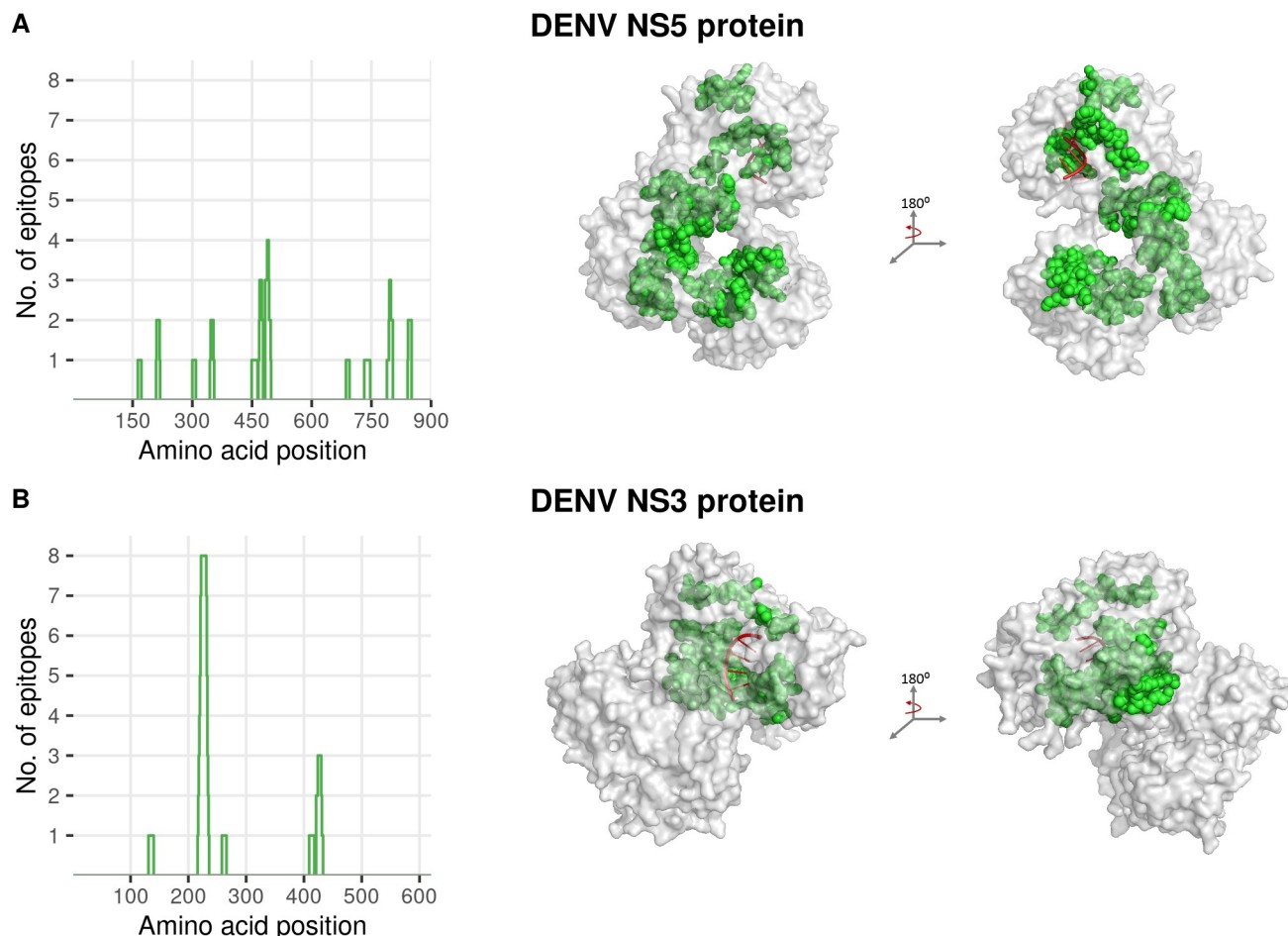

**Fig 4. Mapping of top conserved DENV NS5 and NS3 epitopes onto the protein primary and tertiary structures. (A)** (Left panel) Coverage of the top conserved NS5 epitopes (Fig 3) along the protein sequence, and (right panel) their locations in the corresponding tertiary structure (PDB ID 5DTO). Most of the residues within these epitopes are located in regions that provide stability to the polymerase complex [52]. **(B)** (Left panel) Coverage of the top conserved NS3 epitopes (Fig 3) along the protein sequence, and (right panel) their locations in the corresponding tertiary structure (PDB ID 5XC6) [50]. Although these epitopes are spread out in the primary structure, they localize near the RNA binding groove of the helicase domain [58]. Right panels show side views of the protein structure, rotated by 180°. Residues that belong to conserved epitopes are shown as green spheres while the RNA, interacting with helicase of NS3 (B) and methyltransferase of NS5 (A), is shown in red.

provide biological reasoning, complementary to the statistical analysis, to further suggest that training T cells to recognize the identified top conserved epitopes through vaccination may potentially be helpful in eliciting a robust T cell response against DENV.

### Identifying epitopes that can serve as targets for a universal T cell-based dengue vaccine

We sought to identify a robust set of epitopes as targets for a potential universal T cell-based dengue vaccine that would maximize chances of eliciting a T cell response against the conserved regions of DENV in a large percentage of the population. To address this, we used the global prevalence data of the HLA alleles [46] that are associated with the identified top 55 cross-serotypically conserved DENV epitopes (Fig 3).

We first estimated the population coverage of each of the top epitopes and ranked them accordingly. We then sought to determine sets of epitopes that provided maximum global

population coverage. We started by considering the epitope (LEFEALGFLNEDHWF) that had the highest individual population coverage (52.92%), and then progressively added epitopes such that the accumulated population coverage was maximized (see Methods for details). We continued this procedure until we obtained a set of epitopes for which the accumulated population coverage saturated. This procedure identified a set of 17 epitopes, from the top 55 epitopes (Fig 3), which provided an accumulated global population coverage of 99.23% (Fig 5A). Three epitopes in this set were highly conserved across all DENV serotypes, while the remaining were highly conserved in three of the four serotypes. Of these, DENV2 is covered by the greatest number of epitopes (16), followed by DENV3 (14), DENV4 (13) and DENV1 (11), respectively.

Based on our analysis we recommend these 17 epitopes (8 HLA class I and 9 HLA class II restricted) as potentially robust targets for a universal T cell-based vaccine against DENV. Interestingly, three epitopes within this set (LEFEALGFLNEDHWF, LPAIVREAI, and RVIDPRRCLK) together provided a global population coverage of greater than 80% due to their known associations with multiple distinct HLA alleles [45]. It is encouraging that the proposed set has a high population coverage within individual populations across multiple DENV-endemic countries spread across South-East Asia, Central and South Americas and the Western Pacific [1,5,7] (Fig 5B). Importantly, the estimated population coverage of the proposed set of epitopes is high within most countries that have recently experienced a high incidence of dengue [41], such as Thailand (98.25%), Brazil (98.8%) and Philippines (96.73%). To further demonstrate the applicability of our approach, we also proposed separate sets of epitopes as potentially robust targets for vaccines specifically targeting populations within these dengue-endemic countries by identifying the smallest sets of epitopes that achieved the maximum population coverages. These sets comprised 12, 13 and 5 epitopes, respectively (S7 Fig). With the exception of two epitopes (KGSRAIWYMW and YLAGAGLAF) in the Thailand specific set, and one epitope (RTLILAPTRVVAAEM) in the Brazil-specific set, all the epitopes within the three country-specific target sets are also part of our proposed set for a universal dengue vaccine (Fig 5).

## Discussion

Incidence of infections from all serotypes of DENV continue to rise across the world, affecting more countries than ever before. Despite various efforts there is still no safe and effective vaccine that protects against dengue. Increasing experimental evidence indicates the potential protective role of T cell-based immune responses against DENV [13]. As all four serotypes of DENV are known to co-circulate in endemic regions and with increased risk of severe dengue from heterotypic infections, an effective vaccine should elicit protective T cell responses against most, if not all, serotypes. A fundamental question in designing such a cross-serotypic vaccine is whether there exist parts of the DENV proteome that are: (i) highly conserved across a majority of the four DENV serotypes, and (ii) immunogenic so that T cells can be trained to recognize them. In seeking to address this question, we analyzed the conservation profile of all experimentally-determined DENV T cell epitopes derived from human hosts within each serotype. We identified 55 T cell epitopes that are highly conserved within at least three serotypes. Their high conservation is apparently driven by functional/structural constraints imposed by the corresponding residues. Training T cells against these epitopes is expected to elicit robust responses against at least three DENV serotypes which may potentially be difficult to escape. Based on the identified top conserved epitopes, we also proposed a set of epitopes as recommended targets for a universal T cell-based dengue vaccine. This was chosen to maximize the global population coverage using HLA information associated with each epitope.

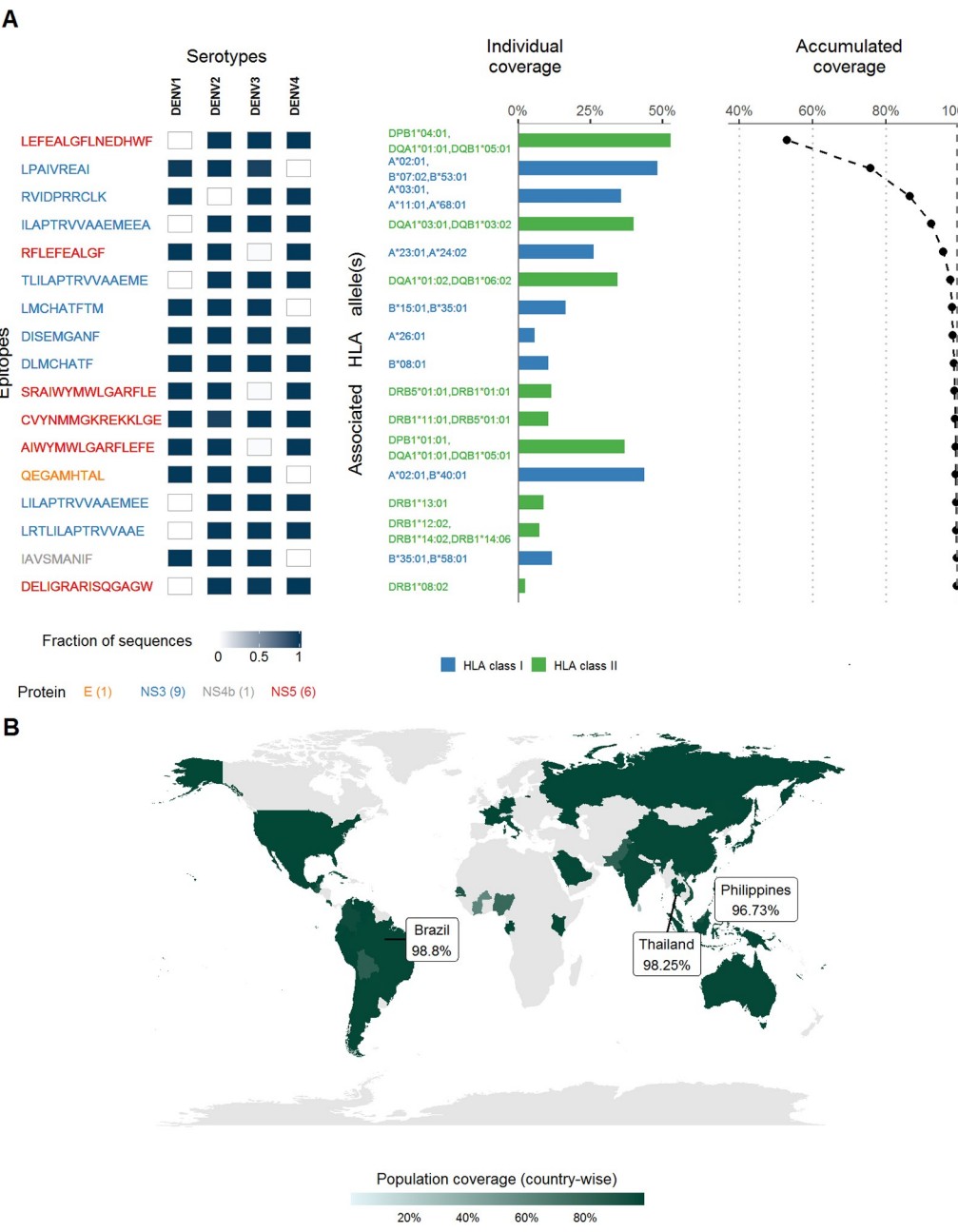

**Fig 5. Candidate for a universal DENV T-cell-based vaccine. (A)** Proposed set comprising 17 epitopes, selected from the set of top 55 DENV epitopes (Fig 3), that maximizes the global population coverage (left panel). The cells adjacent to each epitope represent its conservation within each DENV serotype and are colored according to the scale shown at the bottom. The population coverage of each epitope individually (middle panel) and the accumulated population coverage (right panel) of all epitopes up to and including the current one was calculated based on the information of the associated HLA alleles. Epitopes are ranked in increasing order of the accumulated coverage, which reached the maximum of 99.23%. Incorporating more epitopes within the set did not result in further increase of coverage (see S6 Fig). Epitopes are colored according to the protein from which they are derived while the HLA alleles are colored according to their class restriction. The number of epitopes within the proposed set derived from each protein is shown within parentheses at the bottom of left panel. **(B)** Population coverage by country of the proposed set of epitopes. Coverage is high in most countries, especially those with high incidence rate of dengue infections [41] such as Thailand (98.25%), Brazil (98.8%) and Philippines (96.73%). Countries for which the population coverage could not be computed due to the lack of country-specific HLA information or if no sequences were reported from them, are colored as grey. The map was generated using the "maps" package (https://cran.r-project.org/web/packages/maps/index.html) through the R programming language.

Most vaccine design efforts rely upon the use of LAVs to train the immune system for defending against DENV. However, LAVs pose multiple concerns. This includes mutation of the vaccine strain to virulent form, and the possibility of leading to severe dengue through ADE upon subsequent infection. In contrast, T cell-based vaccines are safer alternatives [59] as they not only eliminate the risks associated with LAVs, but they also provide the opportunity to explicitly train the immune system to target more precise regions of the viral proteome. Importantly, strong T cell responses against more conserved regions of DENV proteins have been correlated with protection from progression to severe disease [15].

Vaccination strategies targeting specifically the conserved epitopes are expected to develop cross-reactive T cell responses which would be expanded upon subsequent natural infection by any DENV serotype. However, this has not yet been demonstrated and further experimental studies are needed to examine the T cell responses elicited upon challenge with different serotypes following such vaccination. Encouragingly, there is evidence that T cell responses can be elicited against "conserved" epitopes among dengue/flavivirus-naïve individuals vaccinated with monovalent as well as tetravalent LAVs, namely TV003 and TAK-003 vaccines [60,61], which are in advanced stages of clinical trials. Detectable T cell responses targeting some of the top 55 conserved epitopes identified in this study (Fig 3) have been reported in naïve individuals following immunization with these LAVs. Specifically, following vaccination with monovalent or tetravalent LAV TV003 [60] and with tetravalent LAV TAK-003 [61], T cell responses have been reported against 9 and 11 of the 55 identified epitopes, respectively (S4 Table). Moreover, there is evidence that a pre-existing T cell response against conserved epitopes is expanded in secondary infection [27,31]. Therefore, focusing the T cell response towards cross-serotypically conserved regions and away from serotype-specific regions, as proposed in this work, through properly designed T cell-based vaccines appears to be a promising vaccination strategy against DENV.

Various vaccine platforms (e.g., based on Modified Vaccinia Ankara (MVA) viral vector, polypeptide mixtures, fusion proteins and nanoparticles) which have been studied in clinical trials for T cell-based influenza vaccines [62–64] may be utilized for designing prospective T cell-based dengue vaccines employing the set of epitopes identified in this work. However, the T cell responses elicited against the combined set of (17) recommended epitopes need to be examined experimentally to determine their strength and the resulting immunodominance pattern of the epitopes. This is important because the sequence context in which the epitopes are delivered influences the intra-cellular processing and presentation of the T cell epitopes [65,66] and affects their immunodominance hierarchy [27]. Specifically, DENV epitopes are reported to have different immunodominance hierarchies depending upon the serotype as well as the order (primary vs secondary) of natural infection [27]. Furthermore, the presence of other antigenic peptides (i.e., potential epitopes) with respect to the cognate HLA molecules and the peptide-HLA binding affinity also affects epitope immunodominance [31,67]. Thus, while immunodominant epitopes are likely to elicit strong responses, precisely determining the immunodominance hierarchy of T cell epitopes is complex and beyond the scope of this study. This would require carefully designed experimental studies to examine the immune responses elicited following immunization with the proposed set of identified epitopes upon challenge with each DENV serotype.

There is evidence that by priming through vaccination, the immunodominance hierarchy can be altered and the immune response can be directed towards otherwise weakly immunodominant epitopes [66,68,69]. In fact, targeting such weakly immunodominant epitopes has been shown to even confer protection [23,70,71]. Thus, while a few of the 55 identified cross-serotypically conserved epitopes appear to be weakly immunodominant (Fig 3), we considered all of them when finding the subset of epitopes for recommending as vaccine targets (Fig 5).

The cross-serotypically conserved epitopes identified in our work differ from those obtained in previous related works [35–37]. Specifically, 44 pan-DENV conserved peptides (9 to 22 residues long) were identified in [35], 78 conserved regions (9 to 139 residues long) were identified in [36], and 46 peptides (15 to 18 residues long) were identified in [37]. None of these matched exactly with the 55 top conserved epitopes identified in this work. We note, however, that some of the conserved regions/peptides reported in the previous studies (10 peptides from [35] and 12 peptides from [37]), overlap partially with the top conserved epitopes identified in this work (S5 Table). We identify three main reasons for these differences. First, there is a larger amount of immunological and sequence data available now compared with past years when the previous studies were conducted, and this would affect the conservation profile of the epitopes. Second, our procedure is epitope-centric, which focuses on the conservation of precise DENV-specific T cell epitopes which have been experimentally-determined from human hosts, while the previous works [35,36] predicted conserved (large) regions/peptides from the sequence data alone which may not represent the precise epitope sequences processed and presented by the HLA molecules, and may not be immunogenic. Third, our definition of epitope conservation based on an "exact mapping" criterion is stricter and potentially more robust than the criteria used in previous works. This strict criterion is important for identifying robust vaccine targets because T cells trained against epitopes may not cross-react with even single residue variants of those epitopes (e.g., as reported for DENV epitope variants EENM**D**VEIW and EENM**E**VEIW [25]).

While we have recommended a set of target epitopes for a prospective T cell-based dengue vaccine that seeks to target a large proportion of the global population, the proposed framework also provides a template for identifying robust vaccine target recommendations for any specific population. The only additional information required is the HLA distribution of the specific population being targeted, which can be seamlessly incorporated with the accompanying source code (see https://github.com/faraz107/Robust-DENV-Vaccine-Candidates). We demonstrated this by identifying country-specific sets that maximize the individual population coverages for three dengue-endemic countries. This shows the promise of the proposed approach in identifying sets of epitopes for prospective vaccines targeting specific populations that are most adversely affected by dengue. Moreover, the developed framework can also be extended to perform a systematic study for identifying conserved epitopes across the flavivirus genus, particularly since there is evidence of cross-reactive T cell epitopes across different flaviviruses [72,73]. Similar approaches have been employed recently to identify potential immune targets for viruses belonging to other genera, e.g., SARS-CoV-2, the virus causing the ongoing COVID-19 pandemic [74–76].

We have also provided a comprehensive list of conservation profiles of 1,768 experimentally-determined DENV T cell epitopes with their associated HLAs (S1 File) which could serve as a reference for future studies. This comprehensive list can aid in reinterpreting earlier results. For example, the epitope NIQTAINQV was previously understood as being conserved across multiple DENV serotypes [77]. However, using the provided list one can observe that this epitope is only conserved in DENV2. Furthermore, it would be interesting to investigate the functional/structural importance of all the top cross-serotypically conserved epitopes identified in this work to better understand molecular determinants of their conservation.

## Supporting information

**S1 Fig. Coverage of DENV T cell epitopes across the primary structure of DENV proteins.** The locations of epitopes were determined by mapping them onto all: (A) DENV2, (B) DENV3, and (C) DENV4 sequences, respectively. The color scales in (A)-(C) indicate the

HLA class restriction of the epitopes.
(PDF)

**S2 Fig. Top conserved epitopes are different only at DENV serotype-specific residues.**
Sequence-logos of top epitopes that are not highly conserved in one of the four DENV sero-
types for: **(A)** 15 epitopes of NS5, **(B)** 16 epitopes of NS3, and **(C)** 2 epitopes of E (top panel)
and 3 epitopes of NS4b (bottom panel).
(PDF)

**S3 Fig. Conservation of the 1,768 T cell epitope sequences across the DENV serotypes.**
Cells adjacent to each epitope represents its conservation within each DENV serotype. The
conservation level (i.e., fraction of sequences in which a given epitope was exactly mapped) for
an epitope within a serotype was determined by mapping it onto all the corresponding protein
sequences for that serotype, as shown in Fig 2B. All epitopes are shown here in descending
order (top to bottom then left to right) of their mean conservation across the serotypes and
colored according to the protein from which they are derived.
(PDF)

**S4 Fig. Histogram of minimum conservation of epitopes among 3 DENV serotypes.** Mini-
mum conservation of epitopes across sets of 3 out of 4 DENV serotypes. The conservation for
each epitope within each serotype of DENV was determined after mapping the epitopes onto
the corresponding protein sequences, as shown in Fig 2B. A threshold of 0.9 was used that
resulted in the set of 55 top epitopes (i.e., epitopes that mapped exactly onto at least 90% of
sequences in at least 3 of the 4 DENV serotypes were selected), shown in Fig 3. The histograms
show that the selection of epitopes is robust to the choice of threshold.
(PDF)

**S5 Fig. Some top epitopes are putatively involved in the inter-dimer interface in the qua-
ternary structure proposed for DENV NS5.** The NS5 dimer structure (PDB: 5CCV) proposed
in [53] is shown where one monomer is colored blue and the other green. The residues of two
top NS5 epitopes (GPGHEEPIPM and KVRKDIPQW) that are involved in the inter-dimer
interface are shown as spheres.
(PDF)

**S6 Fig. Global population coverage for the identified top DENV T cell epitopes.** (Left
panel) Top 51 epitopes, selected from the set of top 55 DENV epitopes (Fig 3) that had at least
one HLA allele associated with 4-digit resolution. The cells adjacent to each epitope represent
its conservation within each DENV serotype. The individual population coverage of each epi-
tope (Middle panel) and the accumulated population coverage of the combination of epitopes
(Right panel) was calculated based on the associated HLA alleles. The epitopes are ranked in
increasing order of the accumulated coverage which reached the maximum of 99.23% with top
17 epitopes. The remaining epitopes are ordered in decreasing order of their mean conserva-
tion (Fig 3). Epitopes are colored according to the protein from which they are derived while
the HLA alleles are colored according to their class restriction. Number of epitopes derived
from each protein is shown within parentheses at the bottom of left panel.
(PDF)

**S7 Fig. Proposed immunogens for country-specific populations. (A)** Thailand, **(B)** Philip-
pines and **(C)** Brazil. For all (A-C): (Left panel) Proposed immunogens comprising epitopes
selected from the set of top 55 DENV epitopes (Fig 3), that maximized the country-specific
population coverages. The cells adjacent to each epitope represent its conservation within each
DENV serotype. (Middle panel) The individual population coverage of each epitope and (right

panel) the accumulated population coverage calculated based on the associated HLA alleles. The epitopes are ranked in increasing order of the accumulated coverage which reached the maximum of: (A) 98.25%, (B) 96.73%, and (C) 98.8%. Incorporating more top epitopes within these immunogens did not result in further increase of coverage. Epitopes are colored according to the protein from which they are derived while the HLA alleles are colored according to their class restriction. Number of epitopes within the immunogens derived from each protein is shown within parentheses at the bottom of left panel in (A-C) respectively.
(PDF)

**S8 Fig. Reported epitope-HLA binding affinity for the top 55 conserved epitopes (Fig 3) in the context of cognate HLA alleles.** The quantitative measure, IC50 values, of epitope-MHC binding was extracted from positive MHC assays corresponding to these epitopes in the IEDB database. Color of the cells represent the strength of the binding affinity. White cells represent epitope-HLA complexes for which no binding measurements were reported.
(PDF)

**S9 Fig. Cross-serotypically-conserved DENV T cell epitopes from HLA-transgenic mice.** Of the 213 experimentally-determined HLA-transgenic mice epitopes, 9 mapped exactly onto at least 90% of sequences in at least 3 of the 4 DENV serotypes. Five of these epitopes–APTRV-VAAEM, ELMRRGDLPV, RVIDPRRCL, SRAIWYMWLGARFLE, and LPAIVREAI–were also reported from human hosts (Fig 3). Cells adjacent to each epitope represents its conservation within each DENV serotype. The conservation of an epitope within a serotype was determined using the exact mapping procedure (Fig 2B). All epitopes are shown here in descending order (top to bottom) of their mean conservation across the serotypes and colored according to the protein from which they are derived.
(PDF)

**S1 Table. List of ViPR database based strain names and mature peptide IDs of DENV protein sequences used in the analysis.** The 56,496 DENV protein sequences were downloaded from ViPR database [42] (https://www.viprbrc.org/; accessed July 30, 2019).
(CSV)

**S2 Table. List of DENV-specific T cell epitopes used in the analysis.** The 1,768 DENV-specific epitope sequences were downloaded from ViPR database [42] (https://www.viprbrc.org/; accessed November 15, 2019).
(CSV)

**S3 Table. Cross-serotypically conserved epitopes identified in this study (Fig 3) that are reported as immunodominant in previous studies.**
(PDF)

**S4 Table. Cross-serotypically conserved epitopes identified in this study (Fig 3) against which T cell responses were reported in naïve individuals following vaccination.**
(PDF)

**S5 Table. Conserved peptides/epitopes reported in previous works that overlap with cross-serotypically conserved epitopes identified in this study (Fig 3).** Overlapping residues are indicated as red.
(PDF)

**S1 File. Cross-serotypic conservation profiles of DENV T cell epitopes experimentally-determined from human hosts.** This HTML table serves as an easy-to-use tool for browsing, filtering, exploring and exporting conservation profiles and associated HLA alleles information

about DENV T cell epitopes.
(HTML)

**S2 File. Cross-serotypic conservation profiles and immune response data of the top 55 conserved DENV T cell epitopes identified in this study (Fig 3).** Immune response data includes response frequency, and qualitative and quantitative binding reported in IEDB for each identified epitope. The response frequency percentile for each epitope, that we computed using the response frequencies of all available DENV epitopes, is also included in this table.
(HTML)

**S3 File. Cross-serotypic conservation profiles of 213 DENV T cell epitopes experimentally-determined from HLA-transgenic mice.**
(HTML)

## Author Contributions

**Conceptualization:** Syed Faraz Ahmed, Ahmed A. Quadeer, Matthew R. McKay.

**Data curation:** Syed Faraz Ahmed.

**Formal analysis:** Syed Faraz Ahmed, Ahmed A. Quadeer, John P. Barton, Matthew R. McKay.

**Funding acquisition:** Matthew R. McKay.

**Investigation:** Syed Faraz Ahmed, Ahmed A. Quadeer, John P. Barton, Matthew R. McKay.

**Methodology:** Syed Faraz Ahmed, Ahmed A. Quadeer, Matthew R. McKay.

**Project administration:** Matthew R. McKay.

**Resources:** Syed Faraz Ahmed.

**Software:** Syed Faraz Ahmed.

**Supervision:** Ahmed A. Quadeer, John P. Barton, Matthew R. McKay.

**Validation:** Syed Faraz Ahmed, Ahmed A. Quadeer, John P. Barton, Matthew R. McKay.

**Visualization:** Syed Faraz Ahmed.

**Writing – original draft:** Syed Faraz Ahmed, Ahmed A. Quadeer, Matthew R. McKay.

**Writing – review & editing:** Syed Faraz Ahmed, Ahmed A. Quadeer, John P. Barton, Matthew R. McKay.

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
