## [Decision Letter · Decision Letter 0]

2 Jun 2020

Dear Prof. McKay,

Thank you very much for submitting your manuscript "Robust candidates for a universal T cell vaccine against Dengue virus" for consideration at PLOS Neglected Tropical Diseases. As with all papers reviewed by the journal, your manuscript was reviewed by members of the editorial board and by several independent reviewers. In light of the reviews (below this email), we would like to invite the resubmission of a significantly-revised version that takes into account the reviewers' comments. 

We cannot make any decision about publication until we have seen the revised manuscript and your response to the reviewers' comments. Your revised manuscript is also likely to be sent to reviewers for further evaluation.

Sincerely,

Daniela Weiskopf, Ph.D

Guest Editor

Scott Halstead

Deputy Editor

Reviewer's Responses to Questions

**Key Review Criteria Required for Acceptance?**

**Methods**

-Are the objectives of the study clearly articulated with a clear testable hypothesis stated?

-Is the study design appropriate to address the stated objectives?

-Is the population clearly described and appropriate for the hypothesis being tested?

-Is the sample size sufficient to ensure adequate power to address the hypothesis being tested?

-Were correct statistical analysis used to support conclusions?

-Are there concerns about ethical or regulatory requirements being met?

Reviewer #1: This is a computational analysis and the authors have clearly described their methodology for reproducibility purposes.

Reviewer #2: (No Response)

Reviewer #3: - The objectives of the study are clearly specified, although they remain hypothetical and difficult to test.

- The population has been clearly described

- The sample size is sufficient, although some eptides mentioned in the litterature are missing (see comments attached)

- Statistics are correct

- No concerns about ethical or regulatory requirements

**Results**

-Does the analysis presented match the analysis plan?

-Are the results clearly and completely presented?

-Are the figures (Tables, Images) of sufficient quality for clarity?

Reviewer #1: Yes, the results are presented nicely and adhered to the publication standards.

Reviewer #2: (No Response)

Reviewer #3: - The analysis presented match the analysis plan

- The results are clearly and completely presented

- The figures are of sufficient quality

**Conclusions**

-Are the conclusions supported by the data presented?

-Are the limitations of analysis clearly described?

-Do the authors discuss how these data can be helpful to advance our understanding of the topic under study?

-Is public health relevance addressed?

Reviewer #1: The conclusions drawn supported by the results.

Reviewer #2: (No Response)

Reviewer #3: - The conclusion of the identification for a set of candidate-epitopes for a pan DENV T cell-based vaccine, would require more functional studies to verify the immunogenicity against different DENV serotypes

- The limitation of this analysis is not enough clearly stated

- With this limitation, the public health relevance is addressed

**Editorial and Data Presentation Modifications?**

Reviewer #1: (No Response)

Reviewer #2: (No Response)

Reviewer #3: - In Fig. 3 and Fig. 5, strong and weak epitopes should be mentioned. This is an important data to keep in mind, for prospective efficient T cell vaccines.

**Summary and General Comments**

Reviewer #1: The manuscript is an attempt to analyze the DENV data from ViPR database. The authors compared the sequences and epitopes and suggested conserved patterns that are also recognized as epitopes. The manuscript is quite interesting and performed in a very reproducible manner. Though there are certain comments need to be addressed before recommending the manuscript for publication. 

1. The authors used the consensus sequences as standard, however these consensus sequences are not necessarily representing a serotype. As they considered the highest occurring residues at any positions, these residues will be coming from different variants of the same serotype and might not be a part of any individual strain. 

2. The consideration of a consensus sequence for conservation has diluted the idea of evolution. The variations among the different members of serotype are evolved in different geographical regions and therefore each variation carries an information for epitopes escaped or developed in that region. The same is reflected in Figure 1B as the sequences are retrieved from different geographical regions. 

3. The authors did a reverse of geographical regions while calculating the population coverage. They considered the world population as a whole and summed up the frequencies of individuals carrying certain alleles while reporting the population coverage. 

In such a case, if we are designing a vaccine candidate from the consensus sequences, which are dominated in the Asian countries, but taking the population coverage for an allele mostly reported in the European countries.

4. The authors' justification of that their work is quite different from previous studies is not quite strong. 

4.1. The larger dataset not necessarily that results are different too. 

4.2. The stricter criteria in this manuscript might be covered in not that stricter criteria in previous studies. 

Please provide a fair comparison with previous results.

Reviewer #2: (No Response)

Reviewer #3: A more complete discussion, on immunodominant and subdominant epitopes in the context of different protein sequences would reinforce the conclusion of a vaccine-candidate epitope. The question of the hierarchy of epitopes in a whole viral sequence vs a short mosaic sequence is a crucial issue, and should be discussed in more detail.

PLOS authors have the option to publish the peer review history of their article (what does this mean?). If published, this will include your full peer review and any attached files.

Reviewer #1: No

Reviewer #2: No

Reviewer #3: No
---

## [Editor Report · Decision Letter 1]

4 Aug 2020

Dear Prof. McKay,

We are pleased to inform you that your manuscript 'Cross-serotypically conserved epitope recommendations for a universal T cell-based dengue vaccine' has been provisionally accepted for publication in PLOS Neglected Tropical Diseases.

Best regards,

Daniela Weiskopf, Ph.D

Guest Editor

Scott Halstead

Deputy Editor

---

## [Editor Report · Acceptance letter]

11 Sep 2020

Dear Prof. McKay,

We are delighted to inform you that your manuscript, "Cross-serotypically conserved epitope recommendations for a universal T cell-based dengue vaccine," has been formally accepted for publication in PLOS Neglected Tropical Diseases.

Best regards,

Shaden Kamhawi

co-Editor-in-Chief

Paul Brindley

co-Editor-in-Chief
